# A Comprehensive Study of the Microbiome, Resistome, and Physical and Chemical Characteristics of Chicken Waste from Intensive Farms

**DOI:** 10.3390/biom12081132

**Published:** 2022-08-17

**Authors:** Aleksandra Błażejewska, Magdalena Zalewska, Anna Grudniak, Magdalena Popowska

**Affiliations:** 1Department of Bacterial Physiology, Institute of Microbiology, Faculty of Biology, University of Warsaw, Miecznikowa 1, 02-096 Warsaw, Poland; 2Department of Bacterial Genetics, Institute of Microbiology, Faculty of Biology, University of Warsaw, Miecznikowa 1, 02-096 Warsaw, Poland

**Keywords:** antibiotic-resistance genes, chicken waste, intensive farming, Poland

## Abstract

The application of chicken waste to farmland could be detrimental to public health. It may contribute to the dissemination of antibiotic-resistance genes (ARGs) and antibiotic-resistant bacteria (ARB) from feces and their subsequent entry into the food chain. The present study analyzes the metagenome and resistome of chicken manure and litter obtained from a commercial chicken farm in Poland. ARB were isolated, identified, and screened for antibiogram fingerprints using standard microbiological and molecular methods. The physicochemical properties of the chicken waste were also determined. ARGs, integrons, and mobile genetic elements (MGE) in chicken waste were analyzed using high-throughput SmartChip qPCR. The results confirm the presence of many ARGs, probably located in MGE, which can be transferred to other bacteria. Potentially pathogenic or opportunistic microorganisms and phytopathogens were isolated. More than 50% of the isolated strains were classified as being multi-drug resistant, and the remainder were resistant to at least one antibiotic class; these pose a real risk of entering the groundwater and contaminating the surrounding environment. Our results indicate that while chicken manure can be sufficient sources of the nutrients essential for plant growth, its microbiological aspects make this material highly dangerous to the environment.

## 1. Introduction

Chicken meat production is considered one of the fastest growing sectors of the Polish food industry. The US Department of Agriculture (USDA) reports that Poland is one of the largest poultry meat producers in the European Union (EU), with chicken production reaching 2.2 million metric tons in 2020 (USDA, 2020). Total chicken meat production in the EU peaked in 2018, reaching 15.2 million tons; this year, Poland produced almost 17% of this value (176.7 mln chickens, including 192.1 mln laying hens), giving it a leading position within the European Union (Eurostat) [1]. 

A number of public and environmental health organizations, such as the World Health Organization (WHO), have raised concerns regarding the spread of antimicrobial resistance (AMR) in bacteria, driven by the extensive use of antibiotics in poultry and livestock production [2]. The most commonly used antimicrobials in the poultry industry are polymyxins, penicillins, tetracyclines, and trimethoprim with sulfonamides. Antimicrobials such as fluoroquinolones, third- and fourth-generation cephalosporins, macrolides, glycopeptides, and polymyxins are considered by the WHO as the “highest priority critically-important antibiotics for human medicine” due to the limited availability of alternatives for the treatment of bacterial infections. While many of these are approved for use in poultry in the largest poultry-producing countries, fluoroquinolones are banned in the United States, and cephalosporins in the EU [3]. 

Antibiotics are administered to treat intestinal infections such as colibacillosis, necrotic enteritis, and other diseases generally caused by *Salmonella*, *Escherichia coli*, or *Clostridium* spp. These infections are a major concern among poultry producers, leading to enormous economic losses [3]. However, up to 70% of the administered antimicrobial compound is excreted with feces and urine in their active form, and remains in fertilized soil for a prolonged time [4,5]. Furthermore, antibiotics are generally administered to the entire flock, not only to isolated infected birds, with the aim of treating disease (therapy), preventing infection (metaphylaxis), and promoting growth [6]. The birds are, in most cases, kept in high numbers in intensive breeding facilities, and are often transported at high densities over long distances, which makes them especially prone to disease. The type and extent of antibiotics vary between countries based on their economy, level of development, intensity of animal husbandry, and animal species [7]. In addition, the method of administration and the volume of antibiotics used vary according to the stage of production and the risk of disease [8].

Antibiotic residues are considered an important environmental contaminant. They are believed to play a key role in enriching soil microbiomes with antibiotic-resistant bacteria (ARB) and antibiotic-resistance genes (ARGs) by creating on-site pressure, which facilitates the development of new ARGs and supports the transfer of existing ones between bacteria [9,10,11]. 

In Poland, 86.8% of chickens are raised in the cage bedding system (a source of manure), 9.6% in the bedding system (a source of chicken litter), 3.2% in the free-range system, and 0.3% in the organic system. A similar trend can be seen across Europe, with cage bedding systems dominating (58.7%); however, a higher percentage of chickens are raised in the bedding system (27.5%), free-range system (9.2%), and organic system (4.6%) [12]. While farm caged layers and broiler breeders produce chicken manure consisting of only fecal droppings (undigested food, urea, gastrointestinal microbiota), the litter from floor-raised chickens consists primarily of animal droppings (feces with gastrointestinal microbiota) mixed with bedding material such as sawdust, and smaller amounts of feathers and dropped fodder [13]. Some studies have shown that chicken litter contains a complex and dynamic microbiota composed primarily of gastrointestinal tract and environmental bacteria, depending on the litter management regimens [14]. It is estimated that 1000 broilers can produce approximately 120 kg of feces per year [15]. In Poland, the total manure produced by chicken farms in 2017 was estimated to be approximately 4,494,639 tones; hence, the storage and utilization of chicken waste can present a huge challenge for the whole industry. In developing countries, waste is most commonly managed by spreading unprocessed manure and litter on the soil, with or without prior dilution. Indeed, such natural fertilizer application can significantly improve soil properties and fertility, as chicken waste is rich in nitrogen and contains high quantities of phosphorus and potassium, which are essential for plant growth [12]. 

However, despite the clear advantages of animal waste application, this practice has unavoidable risks. The contamination of the soil microbiome, one of the richest and most diverse environments, with antibiotics and antibiotic residues may contribute to the spread of antimicrobial resistance [13]; with the antibiotics, growth hormones, and potential pathogens living in the chicken intestine playing a significant role [16]. Usually, chicken waste is inhabited by diverse microbes, especially Gram-negative bacteria, some of which are potential human pathogens (*E. coli*, *Mycobacterium* spp., *Salmonella* spp.). As these microbes can harbor high levels of ARGs and ARB, the widespread application of untreated waste in the field can present a hazard to the environment [17]. Chicken waste has also been identified as a major reservoir of mobile genetic elements (MGE) facilitating the exchange of ARGs between bacteria via horizontal gene transfer (HGT) [18]. When resistance genes are located on plasmids, transposons or integrons can spread among even non-related bacteria [10]. Recent studies report that phylogeny is one of the main drivers for transferring mobile ARGs in animal and human gut microbiota, and that these elements can also be harbored by human pathogens [19]. 

Fertilizer consisting of chicken manure is characterized by a relatively high prevalence of ARGs, reaching up to 60% in isolated bacteria. More importantly, most of these are classified as being multi-drug resistant (MDR), being resistant to more than three classes of antimicrobials [20]. There is hence a need for novel legal regulations regarding animal waste treatment, and this requires a clearer understanding of the impact of the field application of chicken waste on soil ARB and ARG levels, and their dissemination. Such field applications may significantly impact on the soil resistome, altering natural soil microbial communities.

Our study addresses an important issue—how antimicrobial misuse or overuse affects human health. This topic is inseparably linked with the WHO’s One Health initiative, which is intended to address problems facing human, animal, and environmental health by coordinating researchers from multiple disciplines, including public health practitioners and clinicians, at local, national, and global levels [21]. AMR is clearly associated with animal, human, and environmental health. Indeed, animal production plays a key role in the global AMR crises: intensive farming has used a range of antibiotics in subtherapeutic doses for prolonged periods, creating ideal conditions for the development and transfer of resistance genes between bacteria. These genes can also be transmitted to human pathogens or gut microbiota via people, contaminated food, or the environment [22]. Moreover, the fact that the same antibiotics, or those with a similar chemical composition (i.e., the mode of action), are used in the treatment of humans and animals further drives the spread of resistance among their respective pathogens, either directly or through the environment.

The aim of the present study was to evaluate the potential of chicken waste (manure, litter) as a source of hazardous contaminants released into the natural environment, and as a crucial hot spot for the spread of antimicrobial resistance. Currently, little data exists on the distribution of ARGs and ARB in chicken manure, especially that derived from Poland, one of the main chicken meat producers in the EU. To assess the potential threat, the study analyzed the microbial community composition of two types of chicken waste, viz., chicken manure (CM) and chicken litter (CL), based on sequencing protocols targeting the V3–V4 variable regions of 16S rRNA. The resistome of samples was analyzed using high-throughput qPCR, targeting 384 genes in each sample, to confirm the presence of ARGs. The study also attempted to isolate selected antibiotic-resistant, opportunistic pathogenic bacteria listed by the WHO as the so-called ‘priority pathogens list for R&D of new antibiotics’ from the CL and CM [23].

## 2. Materials and Methods

### 2.1. Manure Samples

Chicken waste samples were collected from two places located in the meat production chain at a single large, commercial, meat-producing farm in Poland (CM—chicken manure from laying hens and CL—chicken litter from broilers). Subsamples of chicken litter were collected from 10 locations (approximately 0.1 kg each) in the chicken house (also around waterers and feeders) and polled together into one representative sample for analysis. Similarly, for laying hens, manure samples were collected from 10 places located under the cages, and pooled together into one representative sample for analysis. All collections and pools were taken in triplicate. The farm owner agreed to manure sampling. After collection, the samples were transported immediately to the laboratory, stored at 4 °C, and processed within 24 h. The samples were subjected to physicochemical analyses at the Wessling Polska Sp. z o.o. (Krakow, Poland) accredited analytical laboratory; these analyses comprised pH value, biogenic element, and heavy metal content: Hg content according to procedure DIN ISO 16772, other heavy metals (ICP-OES/ICP-MS)—DIN EN ISO 11885/DIN EN ISO 17294-2, total nitrogen—VDLUFA, Bd. I, Kap. A 2.2.1, ammoniacal nitrogen—DIN 38406 E5-1 mod., and dry organic matter—DIN EN 12879. Only Hg content and dry organic matter content determination were performed according to the accredited norms: other components were analyzed according to the procedures described in the norms, but adjusted to our needs.

### 2.2. Bacterial Strain Isolation and Characterization

Prior to isolating the microorganisms of interest, the obtained waste was enriched with Brain Heart Infusion broth (for *Enterococcus* spp.) or Luria Bertani broth (for *Klebsiella pneumoniae*, *Escherichia coli*, *Acinetobacter baumannii*, *Pseudomonas aeruginosa*, and *Staphylococcus aureus*) (Table 1) by adding 1 g of chicken waste to 9 mL of liquid medium. The cultures were incubated at 25 °C and 37 °C for 24 h and 48 h, and diluted (10^−1^, 10^−2^, and 10^−3^); and then 100 µL of each dilution was plated on an appropriate selective medium with antibiotic supplementation, according to Table 1. The media and antibiotics were chosen according to the WHO priority pathogens list [23]. Each isolation was performed as three biological replicates, followed by three technical ones. All selective media were purchased from Biomaxima (Lublin, Poland), except for Brillance VRE Agar, which was obtained from OXOID (Thermo Fisher Scientific, Waltham, MA, USA). Antibiotics and inositol were purchased from Sigma-Aldrich (MERCK, Waltham, MA, USA). 

The plates were then incubated according to the recommendations of the medium manufacturer, and again at 25 °C, to select for bacteria originating from the natural environment. Following this, approximately 24–48 colonies resembling the morphology of the intended bacteria were isolated from each medium type, starting from the plate with the highest dilution. Pure cultures were obtained using three consecutive streaking. The resulting cultures were considered pure and stored for further analysis in PBS/glycerol stocks (20% *v*/*v*).

The Kirby–Bauer test for determining antibiotic susceptibility was applied to assess the phenotypic diversity of isolated strains and identify potential clones [24]. Antimicrobial discs (OXOID, MA, USA) were chosen according to the selective medium used for isolation (Appendix A). After incubation for 18 h at 37 °C, the inhibition zones were measured. Strains with different antimicrobial susceptibility profiles (the inhibition zone varied by approximately +/− 2 mm in diameter) around at least one antibiotic disk were considered nonrepetitive and chosen for further research. 

The strains isolated from the chicken waste were classified according to species or genus level, using a matrix-assisted laser desorption/ionization system equipped with a time-of-flight mass spectrometer (MALDI-TOF MS/MS) [25]. The analysis was conducted in an accredited external laboratory, ALAB Laboratoria Sp. z o.o., (Warsaw, Poland), according to a standard diagnostic procedure. Identification was performed by aligning the peaks to the best matching reference data; the resulting log score was classified as follows: ≥2.3, highly probable species; between 2.0 and 2.3, certain genus and probable species; between 1.7 and 2.0, probable genus; and <1.7, non-reliable identification.

Where applicable, antibiotic susceptibility testing was performed using the VITEK^®^ 2 Compact System (BioMerieux, Marcy-l’Étoile, France) [26]. The VITEK system is an automated API system equipped with a EUCAST database that can identify the analyzed microorganisms as being sensitive, resistant, or intermediate. Several antimicrobial susceptibility test cards (AST Cards) were used, depending on the identified bacteria species (AST-P643, AST-ST03, AST-P644, AST-N331, and AST-N332). Each susceptibility test provides an accurate susceptibility phenotype profile for almost all of the tested bacterial strains. Based on the European Committee on Antimicrobial Susceptibility Testing (EUCAST) recommendations (EUCAST, Breakpoint tables for interpretation of MICs and zone diameters, 2020), bacterial strains were assigned as resistant (R), sensitive (S), and intermediate (I) to the applied antibiotics. Standard disk diffusion was used for isolates where automatic testing was not provided due to the lack of the EUCAST recommendations for the particular bacterial species/genus. Microorganisms isolated during the study originated in the natural environment; many of them are not listed in EUCAST guidelines. To be certain that the isolated strains are resistant to the applied antibiotics based only on phenotypic characterization, the following criterion was used: the strains were considered to be resistant only when no inhibition zone appeared around the antibiotic discs. In all other cases, the strains were considered sensitive. This is the first use of such a criterion.

### 2.3. DNA Extraction and Metagenomic Sequencing

DNA extraction was performed on 0.5 g of each type of waste with the FastDNA™ SPIN Kit for Feces and the FastPrep Homogenizer (MP Biomedicals, Santa Ana, CA, USA), according to the manufacturer’s instructions. The DNA concentration was determined using a Qubit fluorometer and a dsDNA BR Assay Kit (Thermo Fisher Scientific, Waltham, MA, USA), and purity was determined by measuring the A260/A280 absorbance ratio with a NanoDrop (Thermo Fisher, USA). Only samples with concentrations higher than 200 µg/mL and with a A260/A280 ratio ranging from 1.8 to 2.0 were chosen for analysis. DNA samples were stored at −20 °C for further use. The DNA samples were isolated in triplicate.

The microbial biodiversity of the waste samples was determined by sequencing targeting of the variable V3–V4 regions of the bacterial 16S rRNA gene, using the Illumina platform. Libraries were prepared with the Nextera^®^ XT Library Preparation Kit (Illumina, San Diego, CA, USA) according to the manufacturer’s recommendations. The obtained datasets were analyzed using the Qiime 2 pipeline with the DADA2 option (sequence quality control) and the SILVA ribosomal RNA amplicon database (taxonomy assigned) [27]. Targeted metagenome sequencing was performed on the Illumina MiSeq platform at the Institute of Biochemistry and Biophysics, Polish Academy of Sciences (IBB, Poland). The obtained data were analyzed and visualized using the Microbiome Analyst web tool (https://www.microbiomeanalyst.ca/, accessed on 28 January 2022) [28].

### 2.4. qPCR Analysis of the ARG Profile

In all chicken waste samples, the presence of ARGs, integrons, and MGE were analyzed using a high-throughput SmartChip qPCR system (Resistomap, Finland). Their amounts were calculated as a relative number of copies (gene copy numbers per 16S rRNA gene copy numbers). The qPCR conditions and initial data processing were performed as described previously [29]. All qPCR reactions were performed in three repetitions, and a threshold cycle (Ct) of 27 was used as the detection limit. Only samples with two or three replicates and the parameters mentioned earlier (Ct ≤ 27, efficiency 1.8–2.2) were regarded as positive. Genes and primers are listed in Appendix A. Further, gene names and gene groups were used across the manuscript according to those listed and categorized in this table.

The relative copy number was calculated as follows: Relative gene copy number = 10^(27−Ct)/(10/3)^ [30]. The ARG copy numbers were calculated by normalizing the relative copy numbers per 16S rRNA gene copy numbers.

Spearman’s correlation coefficient (together with all data visualization) was calculated with GraphPad Prism 9.0.0 (Dotmatrics, Boston, MA, USA) to identify the relationships between ARGs, MGEs, and microbial taxa. The analyzed relationships were used to construct networks using Cytoscape. The correlations were considered as being strong and significant when the absolute value of Spearman’s rank |r|> 0.9 and *p* < 0.01 (unless stated otherwise).

## 3. Results and Discussion

### 3.1. Physicochemical Characterization of Chicken Wastes 

The data on the physicochemical properties of the two kinds of tested chicken waste are presented in Table 2. Typical arable soils found in Poland (sandy, with low humus content, soil evaluation class IV, non-fertilized) were also analyzed as a reference. The collected data shows that the chicken manure (CM) was slightly alkaline (pH = 8), while the litter (CL) samples had a pH closer to the soil value (pH = 6.5); i.e., more neutral and closer to healthy and fertile soil. The pH values of our CL samples differed from those obtained by Dumas et al. [31], but pH value is inseparably related to the moisture content, and this may have differed between sample sites; for example, one may have been closer to a water source. The research from 2010 indicates that the application of chicken manure can significantly increase the soil pH [32]. Both manure samples have higher levels of macronutrients, such as magnesium (Mg), calcium (Ca), and phosphorus (P); and heavy metals such as copper (Cu) and zinc (Zn), compared to the soil. Even so, the heavy metal concentrations in the collected samples were below the maximum permissible limits established by the European Parliament and the Council in the Regulation (EC) No. 1069/2009 from 21 October 2009 [33]. Per kilogram of dry matter, organic fertilizer may contain a maximum of 5 mg Cd, 2 mg Hg, 100 mg Cr, 60 mg Ni, and 140 mg Pb. Some heavy metals may be present in supplements intended to improve animal performance (e.g., zinc for enhancing growth) [34], while others may originate from corroded installations (e.g., cadmium or chrome) [35]. In bacteria, heavy metal tolerance is inseparably connected with antibiotic resistance: it may select for resistant bacteria and facilitate the persistence of resistance genes by cross-resistance, even in the absence of antibiotics in the environment, and facilitate the conjugative transfer of ARGs [36]; e.g., copper-resistant strains are more resistant to ampicillin, tetracycline, chloramphenicol, sulfonamides, and penicillin, compared to copper-sensitive strains [37]. Fang et al. [38] describe a plasmid from the IncHI2 incompatibility group that may carry a number of genes coding for tolerance to heavy metals, such as *pco*ABCDRSE (efflux systems to detoxify copper), *ars*CBRH (efflux systems to detoxify arsenic), or *sil*ESRCBAP (efflux systems to detoxify silver), alongside ARGs such as *flor*R (resistance to amphenicols), *arm*A, *aac*-Ib/*aac*-Ib-cr (aminoglycoside resistance), *bla*CTX–M, *bla*CMY, *bla*SHV, *bla*IMP, *bla*VIM (resistance to β-lactams), *oqx*AB, *qnr*A1, *qnr*S1 and *qnr*B2 (resistance to quinolones), or *fos*A3 (resistance to fosfomycin). This plasmid shows a broad host range: it has been found in *E. coli*, *Salmonella enterica*, *K. pneumoniae*, and *Enterobacter cloacae* isolated from humans and chickens. Pal et al. [39] report that almost 17% of the studied bacterial genomes isolated from different environments harbored both ARGs and metal tolerance genes.

Hence, the presence of sub-lethal doses of heavy metals in bacterial living niches, even within permissible limits, may enhance antimicrobial resistance in the environment. The presence of heavy metals in the chicken feces used for fertilization is especially alarming, because when fertilizer is spread into field, heavy metals enter the natural environment, where they may accumulate and facilitate the persistence of antimicrobial resistance.

CL was almost three times richer in organic matter (OM; 60.2%) and two times richer in total nitrogen (2.78%; %wt N) than CM, with only 18.4% OM and 1.22%wt N. At the same time, the CM samples had more ammoniacal nitrogen (3200 mg/kg), than the CL sample with 2400 mg/kg (Table 2). This disproportion may be due to the differences in the composition of the analyzed waste types: chicken litter additionally has bedding material [13]. 

It is hard to compare the physicochemical characteristics of environmental samples with those of other studies, due to the lack of unified methods and sample parameters. Even so, our results, especially those of the CM samples, seem to be comparable to those of other samples collected from similar sites in other big poultry-producing countries. In poultry waste collected from several sampling sites in Southern Brazil (one of the biggest poultry meat exporters), the average dry matter content was 64.3%, the total nitrogen content was 2.2 % (similar to the value of our CL samples), and the pH was also alkaline: most of the samples were higher than pH 7, which is slightly more alkaline than the CL samples collected for our experiments [40]. In addition, previous studies on composting chicken manure in Manjung Region, Malaysia, found a higher total nitrogen content (5.52%) and a much lower pH (6.1) than our CM samples [41]. Our results were similar to those of chicken manure samples from South Africa, with a pH of 7.97 to 6.94 and a total nitrogen content ranging over 1.6−3.2%, depending on the sampling site. In addition, the total phosphorus content in the collected samples was also similar to our results, ranging from 1963 mg/kg to 2644 mg/kg. The high phosphorus levels in poultry manure and litter occur mainly because chickens only utilize a minimal portion of the supplied phosphorus, with the rest being excreted with feces or urine [42].

Our physiochemical analysis also revealed high levels of calcium (Ca) and phosphorus (P) in the chicken feces. Calcium (Ca) and phosphorus (P) are crucial nutrients: they are linked to many biological processes, such as cell proliferation, bone formation, blood clotting, and energy metabolism, and in chickens, disorders in Ca and P homeostasis are associated with a decline in growth and egg-laying performance. The main sources of Ca and P for laying hens are mineral supplements and plant-derived compounds. In addition, to meet the required levels, a highly productive laying hen diet is supplemented with high-quality inorganic phosphates. However, due to their inefficient use of P, livestock and poultry are significant P producers, and thus, they are a major source of P input into the environment. The mechanisms of P homeostasis are strongly conserved and linked to Ca metabolism: the dietary Ca/P ratio has a strong impact on health and performance, and so it must lie within a physiological range [43]. Imbalances in dietary Ca and P can also result in excess P excretion, which can have negative environmental effects when poultry litter is applied as a fertilizer to the soil, causing eutrophication and environmental pollution [44]. Due to the importance of phosphorus and calcium for chicken growth and performance, chicken diets are typically overloaded with these two minerals, to reduce the likelihood of deficiency [45]. 

In addition, a higher level of calcium was noted in feces from broiler chickens (CL) than from laying hens (CM). Typically, laying hens are additionally supplemented with calcium to provide for eggshell formation and yolk production during the laying period: calcium accounts for 40% of the eggshell weight in the form of CaCO_3_ [43]. Our results suggest that the broiler chickens and laying hens used in the study may obtain similar amounts of calcium in their regular diet, but the laying hens use more calcium during everyday performance. However, excess levels of dietary Ca can reduce the performance of chickens and inhibit growth, and reduce weight gain and feed efficiency [46]. 

### 3.2. Bacterial Community Composition

#### 3.2.1. Identification of Bacteria Species and Antibiotic Susceptibility Testing

An initial isolation of bacteria species using selective media supplemented with antibiotics resulted in the collection of 649 strains (bacteria counts for each medium are presented in Table 3).

After eliminating potential clones, approximately 119 isolates were obtained for final identification and antimicrobial susceptibility analysis. The differences in bacterial species composition between the two analyzed types of chicken waste (CM and CL) are shown in Figure 1. A greater diversity of bacterial strains were isolated from CL than those from CM. In the CM, only Gram-negative, opportunistic pathogens such as *Myroides odoratus* (n = 2; 1.7%), *Providencia rettgeri* (n = 3; 2.5%), and *Ochrobactrum intermedium* (n = 1; 0.8%) were isolated. Species such as *Citrobacter freundii* (n = 1; 0.8%), *Proteus mirabilis* (n = 28; 23%), *E. coli* (n = 23; 19%), and *Myroides odoratimimus* (n = 13; 10.9%) were present in both waste types. The CL waste was dominated by Gram-negative bacteria species, such as *Acinetobacter johnsonii* (n = 1; 0.8%), *Klebsiella aerogenes* (n = 3; 2.5%), *Stenotrophomonas maltophilia* (n = 1; 0.8%), and *Serratia marcescens* (n = 5; 4%); however, one Gram-positive bacterium was isolated: *Staphylococcus lentus* (n = 19; 16%). Apart from the pathogenic *S. lentus*, all isolated microorganisms can be classified as opportunistic pathogens [47].

Depending on the identified species, different AST cards were used to test susceptibility to groups of antibiotics. The percentage of isolates that were sensitive, intermediate, or resistant to the selected antibiotics is shown in Appendix A. Among 91 tested strains, 61.5% (n = 56) were resistant to ciprofloxacin, 50.5% (n = 46) to gentamicin, and 25.6% (n = 23) to trimethoprim/sulfamethoxazole. 

Out of 72 tested strains, 16.7% (n = 12) were susceptible to piperacillin/tazobactam, 7% (n = 5) to imipenem, 2.5% (n = 2) to meropenem, 18.1% (n = 13) to amikacin, 54.2% (n = 39) to tobramycin, and 36.4% (n = 26) to tigecycline. In addition, 57 strains were tested for susceptibility to second-, third-, and fourth generation cephalosporins: 36.8% (n = 21) were resistant to cefuroxime, 52.6% (n = 30) to ceftazidime, 48.3% (n = 27) to cefotaxime, 1.7% (n = 1) to cefepime, and 54.4% (n = 31) to cefuroxime-axetil. The Gram-positive strains (19 strains of *Staphylococcus lentus*) were tested against three common antibiotics: 47.9% (n = 9) of the strains were resistant to erythromycin, 79% (n = 15) to clindamycin, and all were resistant to tetracycline (n = 19) (Appendix A).

From both of the waste types, 23 strains of *E. coli*, 13 strains of *M. odoratimimus*, and one strain of *C. freundii* were isolated and identified. All were resistant to more than five antibiotics. From CL, five strains of *S. marcescens* were found to be resistant to at least three antibiotics; three strains of *K. aerogenes* were resistant to amoxicillin/clavulanic acid and cefuroxime; and one strain of *A. johnsonii* was resistant to piperacillin/tazobactam, ciprofloxacin, and piperacillin. From CM, three *P. rettgeri* strains and one *M. odoratus* strain was found to be resistant to six or more antibiotics. The complete phenotypes of the isolated and identified strains are presented in Table 4. 

Among the samples, *M. odoratimimus* was the most prevalent of the isolated bacteria. *Myroides* spp. are common residents in natural environments such as water and soil. However, these Gram-negative bacilli can carry resistance genes to multiple antibiotics, and as opportunistic pathogens, they may cause infection in humans, especially in immunosuppressed patients [48]. In 2010, an outbreak of *Myroides* spp. caused several urinary tract infections in Tunisian hospitals. The disc diffusion susceptibility test showed that all isolates were resistant to the β-lactam antibiotic imipenem [49]. Our isolates also showed intermediate resistance to imipenem (intermediate being defined according to the EUCAST recommendations ISO 20776-1). Additionally, our isolates were resistant to ciprofloxacin, gentamicin, or tobramycin. 

*E. coli* is another species that is commonly present in chicken wastes. This coliform is one of the main residents of the chicken intestine, and it can survive in natural environments such as water and soil [50]. It is also considered a marker of environmental contamination with fecal matter [51]. In research from Cameroon, the overall percentage of resistant *E. coli* isolates from the chicken litter was approximately 58.4%. After susceptibility testing, almost 83% of them were resistant to more than three antibiotics and the considered MDR strains. Similar findings were obtained for the phenotypes of the *E. coli* isolates in our study, confirming a high level of MDR among the species, with all 23 isolated *E. coli* being resistant to at least eight antibiotics. In the samples collected in Cameroon, the *E. coli* demonstrated lower levels of resistance to ciprofloxacin (36%) and imipenem (45%) than to ampicillin (91%) and amoxicillin with clavulanic acid (89%) [52]. In our isolated *E. coli* strains, resistance to ciprofloxacin (n = 14; 61.5%) was similar to that of amoxicillin (n = 14; 63.2%), with resistance to imipenem being at a much lower level (n = 2; 7%) in the isolated bacteria. Furthermore, in *E. coli* strains isolated from organic poultry fertilizer in Brazil, most were resistant to more than five tested antimicrobials [53].

Moraru et al. [54] report the presence of fluoroquinolone- and ciprofloxacin-resistant *Enterobacteriaceae* in chicken manure. During our study, the most common form of resistance among the isolated strains (n = 91) was to ciprofloxacin (CIP) (n = 56; 61.5%) (Appendix A). This broad-spectrum fluoroquinolone is commonly used against Gram-negative bacteria in veterinary contexts to prevent avian colibacillosis.

These findings are significant, especially from the environmental point of view, as chicken litter is commonly used as a natural fertilizer on arable fields. These practices carry a high risk of potential crop contamination; indeed, a case study on gardening crops from Southern Benin confirmed crop contamination by fecal coliforms including *E. coli* after poultry manure fertilization [55]. Moreover, bacteria from *Providencia* spp., considered to be commensals in the gastrointestinal tract, pose a threat to humans and animals as opportunistic photogenes, and can cause severe gastric infections. In our studies, three isolates of *P**. rettgeri* were resistant to at least three tested groups of antibiotics (beta-lactams, fluoroquinolones, sulfonamides, and polymyxins), which allows them to be classified as possessing an MDR phenotype. In total, 19 Gram-positive *S**. lentus* strains were isolated only from CL in the present study, and all were resistant to at least three antibiotic groups. Graham et al. [56] report that some staphylococci carrying resistance to more than three antibiotics are able to survive even a 120 day storage period in a two-walled, roofed shed. The possession of an MDR phenotype makes any bacterium even more dangerous: not only can it limit the potential treatment strategies in case of infection, but environmental bacteria may transfer resistance vertically or horizontally to other bacteria [10]. 

In many countries, e.g., the US, chicken litter is commonly used as a bedding material for young broiler chicks placed in chicken houses. After being placed in commercial chicken houses where litter serves as the bedding material, chicks are exposed to bacteria that can enter the immature gut. As the gastrointestinal tract of young chickens offers weak resistance to colonization, it can be easily colonized by bacteria. Beginning from day one, chicks begin pecking at and consuming litter materials, inoculating their young gastrointestinal tract with bacteria present in the litter. Thus, the litter microbiome may affect the development process of intestinal microbiota and its eventual composition and structure in chickens [13]. Our study showed that chicken litter carries many pathogens that are resistant to commonly used antibiotics, and which may be transferred to young chicks via bedding materials. The colonization of the poultry gastrointestinal tract with ARB can cause the further spread of resistance among other chickens, and eventually enter the human food chain via contaminated meat.

Our research has clear value in human health risk analysis. Our analysis confirmed the presence of opportunistic pathogens in chicken waste, which may cause severe infections in both animals and humans. In many cases, those bacteria are resistant to antimicrobials that are classified by the WHO as being critically important for human medicine, such as aminoglycosides and carbapenems; third-, fourth-, and fifth-generation cephalosporins; macrolides, monobactams, polymyxins or quinolones; or as highly important, such as lincosamides, sulfonamides, or tetracyclines. It has been estimated that approximately 60% of infectious diseases experienced by humans may be caused by zoonotic pathogens that can carry ARGs, which could be further transmitted to humans; this further highlights the importance of the WHO’s One Health initiative. These zoonotic diseases include anthrax, caused by *Bacillus anthracis*, bovine tuberculosis by *Mycobacterium tuberculosis*, brucellosis by *Brucella abortus*, or hemorrhagic colitis by *E. coli*.

#### 3.2.2. Microbial Community Composition

The microbial community composition of the CM and CL samples was determined using targeted sequencing of the variable V3–V4 region of the 16S rRNA gene, and their richness and diversity were analyzed. Among all tested samples, a total number of 21 phyla were identified. In chicken manure, *Bacteroidetes* (41%) were considered the dominant bacterial phyla, followed by *Firmicutes* (17%), *Proteobacteria* (12%), and *Spirochaetes* (11%). In CL, the dominant phyla were *Firmicutes* (43%), followed by *Bacteroidetes* (28%), *Proteobacteria* (19%), and *Actinobacteria* (9%). Microbiome compositions for both types of feces are presented as relative abundance bar plots (Figure 2).

One of the dominant phyla in our samples was *Bacteroidetes*. This group is a very diverse bacterial phylum that can colonize various environments, including soil, water, and most importantly, the gastrointestinal tract. *Bacteroidetes* act as symbionts in the gastrointestinal tract, mainly helping to degrade biopolymers such as polysaccharides. Despite their beneficial contribution to the environment, some members of this phylum are well-known opportunistic or clinically relevant human and animal pathogens (*Myroides* spp., *Sphingobacterium* spp.) or phytopathogens, such as *Flavobacterium johnsoniae* [57]. Due to their importance in depolymerizing or degrading organic substances, their abundance in ecosystems might enhance the rate of organic matter turnover, leading to enhanced CO_2_ emissions [58]. Its abundance also correlates strongly with pH, with *Bacteroidetes* being more abundant in environments with more alkaline pH values (approximately pH = 8), than in those with more neutral pH values (approximately pH = 6.5–7) [59]. This observation was also confirmed in our samples, in which *Bacteroidetes* were more abundant in CM (pH = 8) than in CL (pH = 6.5). 

The most abundant phylum in the CL samples was *Firmicutes*. This observation agrees with those of Borda-Molina et al. [60], which identify *Firmicutes* as the most abundant bacterial phylum in the chicken gastrointestinal tract. However, our results for the CL differed from those presented by Gurmessa et al. [58], who reported the contents to be *Firmicutes* (54.83%), *Actinobacteria* (33.00%), *Bacteriodetes* (6.86%), and *Proteobacteria* (2.90%); however, as with Gurmessa et al. [58], the three most abundant phyla constituted approximately 90% of the overall bacterial community composition. Other studies have found *Firmicutes* and *Bacteroidetes* to represent almost 90% of healthy gut microbiota in chickens [61]. *Firmicutes* are also responsible for the degradation of polysaccharides and butyrate production in the chicken gastrointestinal tract. Previous taxonomic analyses based on 16S rRNA sequence assignment by Zhang et al. [36] identified 16 phyla, with the four most dominant being similar to our present findings. It appears that chicken manure samples are, in general, mainly dominated by these four phyla (*Bacterioidetes*, *Firmicutes*, *Proteobacteria*, and *Actinobacteria*). In another study, fresh manure and composted samples were dominated by *Actinobacteria* (43.3 vs. 32.6%), *Firmicutes* (38.4 vs. 39.1%), *Proteobacteria* (10.4 vs. 13.5%), and *Bacteroidetes* (6.4 vs. 7.5%) [62]. Ziganshin et al. [63] found *Firmicutes* (23–79%) and *Bacteroidetes* (8–44%) to be the most abundant phyla in raw chicken manure. However, environmental conditions have been found to vary between poultry houses, and these differences can have a major effect on microbial dynamics, even at the microscale level; spatial scaling of microbial diversity can exist even on a scale of a few centimeters [14].

#### 3.2.3. Microbial Diversity

The microbial richness and diversity of the tested samples were analyzed with the Chao1 (Figure 3A) and Shannon (Figure 3B) indexes. 

The results for the groups were compared using the nonparametric Mann–Whitney test. Higher diversity (*p* ≤ 0.1) and richness (*p* ≤ 0.1) values were observed for CM samples than for CL samples, but only as a general trend. In addition, the beta-diversity analysis identified some differences in microbial community structure between the two types of chicken waste (*p* ≤ 0.1), but also only at the trend level; the results were visualized using PCoA (principal coordinate analysis) based on the Bray–Curtis dissimilarity matrices (Figure 4). 

Considerable differences in diversity and richness were observed in the collected samples, and this may be due to the fact that α-diversity is strongly influenced by the physicochemical conditions of the waste and environmental factors. Few studies have compared the antimicrobial communities of wet (manure) and dry (litter) chicken waste. The fact that chicken litter is a mixture of poultry manure and bedding likely influences the physiochemical characteristics of the resulting litter, which subsequently affects the bacterial community structure [58]. Our present findings indicate that CL and CM differ most prominently with regard to moisture, pH value, total nitrogen concentration, and dry organic matter content. All these characteristics can affect the richness and diversity of the bacterial community, which in turn are correlated with the moisture level of the sample: in CM, a higher moisture content can increase α-diversity by enhancing the transport of dissolved nutrients required by the bacteria [64]. Dumas et al. [31] report that wet manure contains a greater richness and diversity of bacteria than dry litter. Increased moisture is present in the CL, which allows more types of bacteria to persist. However, due to the correlation between moisture content and the levels of other physiochemical parameters known to play a role in microbial diversity, such as carbon and nitrogen availability, determining the predominant factor contributing to bacterial diversity is difficult [31].

The diversity and richness of the bacterial community is also influenced by the pH value; although little data exists regarding this effect, particularly for chicken manure, studies have noted a tendency towards higher community richness and diversity in soils with neutral pH, and lower richness and diversity indexes in more alkaline (pH > 8) and acidic (pH < 4.5) soils [59]. In our research, CM (pH = 8) had higher richness and diversity indexes (*p* ≤ 0.1; trend level) than CL, with a pH value close to neutral pH (pH = 6); however, our results are in line with those of Lauber et al. [59], who used samples with similar pH values. 

Other differences may result from other factors that are known to play a crucial role in microbial community structure in chicken waste, such as antimicrobial usage, animal diet, nitrogen availability, or the presence of heavy metals. Other studies indicate considerable variation in the relative abundance of the bacterial community between chickens, irrespective of the core microbiota colonization [60]. A possible explanation is that shifts in microbial composition are influenced by initially colonizing the microbiota, diet, and immune system of the host [65]. Our present findings indicate an elevated level of calcium in both CL and CM waste. Such high calcium concentrations in poultry diets may result in a reduction in the bioavailability of other minerals, and thus, deficiencies in phosphorus, zinc, magnesium, and iron. In addition, changes in Ca and P supplementation may alter the composition and activity of the microbial community in the digestive tract of broilers: a high dietary concentration of calcium has been found to increase the pH of crop and ileum contents, but not of the contents of other gastrointestinal tract segments [66]. Furthermore, Ptak et al. [67] report that reductions in Ca and P levels lowered pH and total bacterial count in the caeca; the pH value has also been found to directly affect the composition and enzymatic activity of the bacterial community. However, no dietary treatment was found to affect *Bacteroides* counts. 

The composition of the microbial community of the crop mucosa was also found to be significantly affected by the Ca and P concentrations in the diet; however, no such effect was observed in digesta samples [60], highlighting the fact that studies on the effects of diet on gut homeostasis should include both digesta and mucosa samples. In addition, Witzig et al. [68] report that cecal samples possess a different global bacterial community structure to that of the crop, jejunum, or ileum, irrespective of the methodological approach, and that this depends on the amount of phosphorus available in the diet. This suggests that the daily intake of P and Ca affects the gastrointestinal microbiome of chickens, as well as their metabolism.

#### 3.2.4. High-Throughput Quantitative PCR Analysis

##### Relative Abundance of AMR and MGE Gene Classes in HT-qPCR Arrays

Table 5 presents the mean relative abundances of the analyzed genes for CM and CL, divided into groups according to Appendix A. More detailed data on the relative abundance of selected genes are presented in Appendix A

Among the ARGs found in all CM samples (Table 6), the most abundant were *tet*PB_2 (tetracycline-resistance gene), *tnp*A_2 (MGE), *cmx*A (phenicol-resistance gene), *qac*E∆1_1 (other; MDR phenotype), *tet*M_3, *tet*X (tetracycline-resistance gene), *qac*E∆1_2 (other; MDR phenotype), *tet*M_1, and *sul*2_2, *sul*1_1 (sulfonamide-resistance genes); ranked from highest abundance to lowest. In CM, the most abundant group of antimicrobials were tetracycline-resistance genes, with *tet*PB_1, *tet*M_1, and *tet*X being the most prevalent (Figure 5).

The most abundant genes in CL (Table 7) were *lnu*A_1 (MLSB-resistance genes), *tnp*A_6, *ISEfm1* (MGE), *erm*B (MLSB), *tet*M_3 (tetracycline-resistance gene), *int*I1_1 (integrons), *tnp*A_2, *tnp*A_1 (MGE), *tet*M_1 (tetracycline-resistance gene), and *sul*1_1 (sulfonamide-resistance gene). The most prevalent in CL were genes coding for resistance to antimicrobials from the MLSB (macrolide–lincosamine–streptogramin B) group, with *lnu*A_1, *erm*B, and *erm*X being the most abundant. However, MGE (*tnp*A_6, *ISEfm1*, *tnp*A_2, *tnp*A_1, and *pNI105map-F*; ranked from high to low) were even more prevalent (Figure 5). Although CL appears to be the environment with the highest HGT potential (due to a high amount of MGE genes), CM can also be classified as an environment of special concern from a microbiological point of view: among the 10 most prevalent genes, two are related to an MDR phenotype.

The beta-diversity of the differences in ARGs and MGEs between the two types of waste was visualized using PCoA (principal coordinate analysis) based on Bray–Curtis dissimilarity matrices (Figure 6). Comparisons of the overall distribution patterns of ARGs and MGEs in the samples demonstrated separations between the two groups, i.e., CL and CM, but only at the trend level (*p* < 0.1).

The mean distribution of the ARG groups found in chicken waste (mean value for CL and CM) directly corresponds to the groups of antibiotics applied to the chickens: polymyxins, penicillin, tetracyclines, trimethoprim, and sulfonamides (Figure 7). Additionally, aminoglycosides and antimicrobials belonging to the MLSB group are approved for use in poultry in Poland, and by the largest chicken meat producers worldwide (China, Brazil, and US) [3]. Tetracycline-resistance genes constitute 19% of all detected ARGs, while aminoglycoside-resistance genes constitute 15%; MLSB-resistance genes are 15%, sulfonamide-resistance genes are 9%, and other (polymyxins included) are 9%. Especially concerning should be the fact that approximately 19% of the detected genes belong to MGE, and approximately 7% belong to integrons. This suggests a high potential for antibiotic-resistance transfer, which will be analyzed further. 

A study of the resistome of raw chicken manure by Peng et al. [69] identified 262 ARGs in chicken manure samples. Similar to our present findings, the main ARGs were aminoglycoside-, tetracycline-, and sulfonamide-resistance genes. In addition, chicken manure was found to be potentially the richest source of ARGs and MGEs of all manures of animal origin [62].

Qian et al. [70] found that all ARGs detected in pig and bovine manure samples were also present in chicken manure. Beta-lactams were found to be the most abundant ARGs in bovine manure (14.4–68.2%), while only residual amounts were present in the two other manure types. This is in line with our present results, where the level of beta-lactam abundance was low in CL samples (0.07 relative abundance calculated per 16S rRNA gene copies), and even lower in CM (0.004 relative abundance calculated per 16S rRNA gene copies). As also reported by Qian et al. [70], MGEs comprising *Int*I-1, Tp*614*, IS*613*, and *tnp*A were detected in all our tested manure samples. 

In other studies conducted by Xu et al. [71], chicken manure was found to demonstrate a higher ARG abundance than other examined sheep and cattle manure. In the chicken manure samples, similar to ours, the most abundant ARG was found to be *tet*X, probably due to the broad range of potential hosts. In addition, the *sul*1 and *sul*2 genes were pre-existing and dominant in those samples, and importantly, their numbers considerably decreased during the composting process [71]. 

Out of the 384 tested ARGs and MGEs, 162 ARGs were identified in CM samples and 237 in CL (Appendix A, Figure 8). Core resistome analysis revealed that the CM and CL samples share 141 genes in common, and that CL has 96 unique genes and CM has 21. The CL samples have both a higher actual abundance of ARGs (Figure 5) and the highest diversity of detected genes. Among the most prevalent genes in CL and CM, only *tet*PB_2 exists exclusively in CM; the remainder are common for both groups of chicken waste, but they exhibit lower frequencies (*tnp*A_2, *cmx*A, *qac*E∆1_1, *tet*M_3, *tet*X, *qac*E∆1_2, *tet*M_1, *sul*1_1, *sul*1_2, *lnu*A_1, *tnp*A_6, *ISEfm1*, *erm*B, *int*I1_1, *tnp*A_2, *tnp*A_1, and *tet*M_1). 

Recent studies comparing human, pig, and chicken gut resistomes that detected a total of 166 ARGs identified the highest ARG levels in the chicken samples (156 ARGs). In addition, although the chicken samples demonstrated the greatest overall abundance, their ARGs had the lowest diversity [72]. This high abundance of ARGs in chicken might be connected with a high breeding density and a short, intense growth period [73]. However, several ARGs and MGEs were also found to be present on farms with no history of antibiotic usage; similarly to our present samples, the most prevalent ARGs in samples from the antibiotic-free farms were the aminoglycoside-resistance genes *aad*A, *aad*A1, *aad*A2, and *str*B, the sulfonamide-resistance gene *sul*2, and the tetracycline-resistance genes *tet*M, *tet*K, and *tet*X [64].

##### Correlation Analysis of ARGs, MGEs, and Microbial Communities

Correlation between ARGs and MGEs

An ARG and MGE co-occurrence network based on Spearman’s correlations was constructed to analyze the potential risk of spread of antimicrobial resistance into the environment via HGT (Figure 9). This network was based on 1068 significant correlations (46 negative and 1022 positive) and contained 218 nodes (25 MGEs and 193 ARGs) and 1069 edges. As the size of the nodes is related to the degree of its interactions, our network indicates that MGEs play a crucial role. Among all the detected MGEs, IncW_trwAB and IncQ_oriT (plasmid incompatibility) and *trf*A (transposase) were found to have the largest number of significant interactions with ARGs. In addition, two MGEs were found to have a negative interaction with tetracycline-resistance genes: *tnp*A_3 (transposase) and *pNI105map-F* (plasmid replication)—the former with *tet*R_3, *tet*G_2, and *tet*R_2; and the latter with *tet*M_2, *tet*36_2, and *tet*PA. The only gene identified in the tested samples encoding quinolone-resistance (*qnr*B) only demonstrated positive interactions with nine MGEs (e.g., *int*I3_1, *IncN_rep*). *Sul*1_2 showed a negative correlation with the transposase gene *tnp*A_3, and *sul*1_3 with *tnp*A_7. 

Among the beta-lactam-resistance genes, the *tnp*A_7 (transposase) gene is negatively correlated with three genes: *bla*Z, *bla*OXA1/*bla*OXA30, and *mec*A (methicillin-resistance). Two different genes, viz.,Tp*614* (transposase) and IS*Sm2* (insertional sequence), demonstrate negative interactions with the *bla*TLA and *bla*SHV_3 genes, with the latter having an additional relationship with the *pbp* gene. The only negative interaction between the aminoglycoside-resistance genes and MGEs is that between *aad*A5_2 and *tnp*A_1 (transposase). Furthermore, in this network, a negative correlation can be seen between *tnp*A_3 and two other MGEs (IS*Sm2*—insertional sequence and Tp*614*—transposase). 

Several negative correlations were found between vancomycin-resistance genes and MGEs, such as these encoding the insertional sequence IS*Sm2* and Tp*614*; the former is negatively correlated with two genes *van*SA and *van*RA_1, and the latter with *van*RA_2. Out of the several positive interactions between genes encoding the MLSB phenotype, the *pNI105map-F* gene is negatively correlated with *mph*B (macrolide) and *erm*35 (antibiotic target alteration). Similarly, the *mph*C gene has a negative correlation with *ISSm2* and *Tp614*. Additionally *Tp614* is negatively correlated with *msr*A_1 (ABC-F ATP-binding cassette ribosomal protection protein). The gene *erm*Y (a plasmid-mediated methyltransferase) is negatively correlated with two variants of transposase *tnp*A_1 and *tnp*A_3; the latter also interacts negatively with *mat*A/*mel* genes. For the *cfr* gene (ribosomal RNA methyltransferase), six out of eight significant interactions with MGEs were found to be negative (e.g., Tn*5*, IS*1111*, and *trf*A).

Such a network analysis can play an important role in understanding the role of MGEs in the transfer of antimicrobial resistance to the environment, and can provide new information regarding the possible spreading mechanisms and patterns. One of the most prevalent groups of genes in manure and soil are the MLSB-resistance genes, such as erythromycin ribosome methylation (*erm*) [69,74]. Zhang et al. [11] studied the succession of soil antibiotic resistance following the application of three types of animal manure; the results indicate a significant correlation between three *erm* genes (*erm*A, *erm*B, and *erm*C), with transposons indicating their potential transfer [11]. In our studies, these erm genes were found to interact significantly with the *tnp*A (transposase) gene subtypes. 

Another group of antimicrobials widely used in veterinary scenarios are beta-lactams. They are often applied to treat diseases, such as mastitis, and for prophylaxis in animals (e.g., dry cow therapy used in dairy cattle). Extended-spectrum beta-lactamase (ESBL) producers can be isolated from different environments (soil, water, and animal manure) [75]. Evidence exists supporting the possible spread of ESBL genes via HGT; for example, the *bla*TEM and *bla*CTX-M-1 genes detected in pig manure samples have been associated with IncN plasmids [76,77]. In our network analyses, *bla*CTX-M-1 was found to interact with two types of genes, (I) introns (*int*I1_2, *int*I3_2) and (II) transposases (*tnp*A_3 and *tnp*A_6), and only *bla*TEM interacted with the *IncN_rep* gene. 

Two groups of ARGs known to be prevalent in manures are the tetracycline- and sulfonamides-resistance genes. Due to their low absorbance in the gastric tract, most of the sulfonamide and tetracycline administered to an animal exert pressure for ARB survival in the animal intestine; the extra components are excreted as active compounds with feces and urine into the environment [78]. Hence, resistance genes against those particular antibiotics are present in high amounts in fresh manure samples and manure-treated soils [79,80,81]. Various types of *tet* genes present in animal manure are inserted in transposons [82,83]. Most of the detected *tet* genes in our network are clustered and connected with *Tp614*, IS*614*, and I*SSm2* (insertional sequences) and *tnp*A_3 (transposase) genes. 

Sulfonamide-resistance genes (*sul*1, *sul*2, and *sul*3) are widely present in many natural environments such as soil and animal manure. They can be located on MGEs and they transfer between non-related bacteria via mobilized broad host plasmids [29]. *Sul*1 and *sul*2 are frequently associated with class 1 integrons (*int*I), and hence, they can be easily transferred [84,85]. Our network confirmed a strong and significant association between the *sul* and *int* genes. *Sul*1_1 genes interact only with the *int*I2_2 and *int*I1_4 subtypes. *Sul*1_2 and *qac*EΔ1_*1* interact with the *int*I1_3 gene. Most importantly, *sul*2_1 and *sul*1_2 genes are also connected with Tp*614*, IS*614*, and ISS*m2* (insertional sequences). 

The only gene connected to the IncP_oriT gene (plasmid) is the mercury reductase gene (*mer*A). The presence of this gene was confirmed to be strongly connected with the presence of IncP-α plasmid in high arctic snow, freshwater, and sea-ice brine [86]. Plasmids from this incompatibility group can spread resistance genes among even non-related phylogenetic groups of Gram-negative bacteria [87]. 

2.Correlation between detected genes and microbial taxa at the phylum level

The second network, showing the interactions between nine taxa at phylum level and eight gene classes (seven ARG classes and MGEs encoding genes) in chicken waste, is based on 79 strong and significant correlations (41 positive and 38 negative). The network was formed from17 nodes (eight antibiotic groups and MGEs, and nine microbial taxa at the phylum level). According to the network, organized according to Spearman’s rank analysis (|r|> 0.7, *p* < 0.05), various ARGs appear to be significantly associated with different bacterial phyla. 

The potential host microorganisms that were positively correlated with ARGs mainly belonged to *Actinobacteria*, *Proteobacteria*, and *Firmicutes*. *Bacteroidetes* had significant positive interactions only with sulfonamide- and tetracycline-resistance genes. When it comes to antimicrobial groups, quinolone-resistance genes had a significant positive correlation with only two phyla: *Actinobacteria* and *Firmicutes*. Amphenicol-resistance genes had a negative correlation with almost all taxa except for *Actinobacteria*. Sulfonamide-resistance genes had a negative correlation with *Chloroflexi*, *Bacteroidetes*, and *Tenericutes*. MGEs showed positive correlations with *Proteobacteria*, *Actinobacteria*, and *Firmicutes* (three phyla characterized as being dominant in the examined samples) (Figure 10). Similar effects were presented by Zhou et al. [88], which identified the same potential hosts of ARGs as in our analyses, belonging to the four most dominant phyla identified in the samples (*Firmicutes*, *Proteobacteria*, *Bacteroidetes*, and *Actinobacteria*). 

In other studies conducted on composting chicken manure, most of the positive interactions with ARGs were noted for *Proteobacteria*, e.g., *Pseudomonas*, *Halomonas*, and *Stenotrophomonas*. This group demonstrated a significant correlation with the *tet*O, *tet*C, and *sul*1 genes [89]. In our samples, the ARGs coding resistance to tetracyclines only displayed a significant and positive correlation with Bacteroidetes. In animal manure, more than half of the significant correlations with ARGs and MGEs were attributed to *Proteobacteria* and *Bacteroidetes*; these were also the dominant phyla in our analyzed samples [90]. 

## 4. Conclusions

Our findings indicate that the examined chicken waste can be a sufficient source of nutrients essential for plant growth, such as nitrogen and phosphorus, especially considering that the amounts of heavy metals in the chicken manures were within acceptable levels and would not likely pose an environmental threat when directly applied to the soil. Thus, taking into consideration only the physicochemical parameters of chicken wastes, these types of manure can be applied onto arable fields as a natural fertilizer with high level nutrition potential. The microbial community composition analyses found the dominant bacterial phyla in chicken waste to be *Bacteroidetes*, *Firmicutes*, and *Proteobacteria*. Our results also found the diversity and richness of the microbial communities to be associated with the physical and chemical conditions of the collected samples. 

During the study, we identified an alarming diversity and abundance of ARGs and MGEs, and our data indicate positive correlations between them, suggesting a considerable risk of ARG spread into the environment, and their transfer between bacterial species. The most prevalent antibiotic resistance genes in both types of studied chicken waste, viz., chicken litter (CL) and chicken manure (CM), were aminoglycoside-, sulfonamide-, and tetracycline-resistance genes. In CL, genes related to the MLSB phenotype predominated. Interestingly, CL and CM were found to differ when comparing the overall distribution patterns of antibiotic-resistance genes and mobile genetic elements. 

Moreover, we isolated antibiotic-resistant bacteria strains. Some of them were pathogenic, such as *Staphylococcus lentus*, while others were considered to be opportunistic human and animal pathogens, such as *M. odoratus* or *M. odoratimimus*, or phytopathogens, such *as Flavobacterium johnsoniae*. Even more concerning is that many of them were resistant to at least one antibiotic class, and more than 50% of the isolated strains were classified as MDR, due to resistance against more than three antibiotic classes. Hence, it can be concluded that the opportunistic pathogens with MDR phenotypes present in chicken manure and litter pose a real risk of reaching the ground and groundwater and contaminating the surrounding environment, which poses a potential risk to public health. Therefore, taking into account all of the results obtained, it is fair to qualify the studied chicken waste as being highly dangerous material.

## Figures and Tables

**Figure 1 biomolecules-12-01132-f001:**
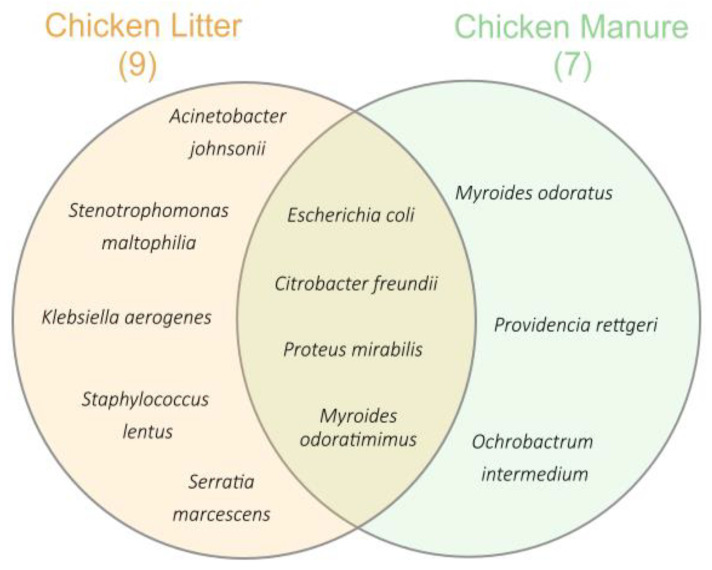
Venn diagram presenting the differences in bacteria species found in two analyzed types of chicken waste—CM and CL; nine bacteria species were detected in CL and seven in CM.

**Figure 2 biomolecules-12-01132-f002:**
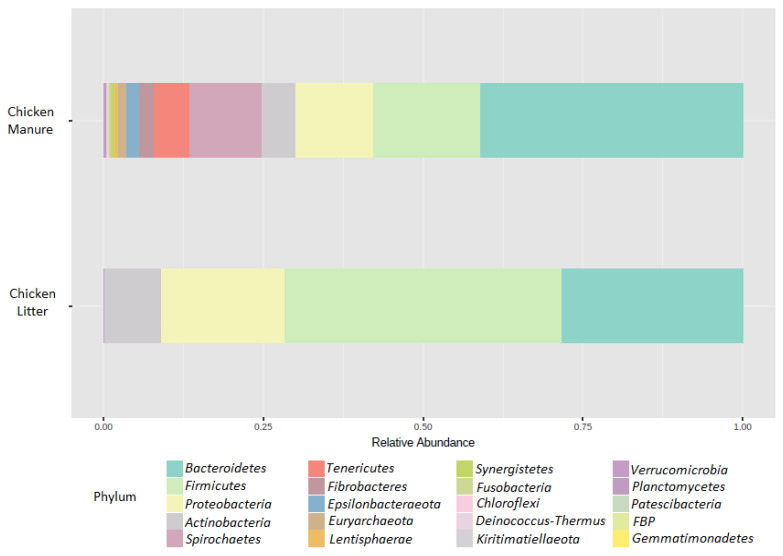
Microbial composition of collected feces samples at the phylum level. The visualization represents the mean value for three replicates.

**Figure 3 biomolecules-12-01132-f003:**
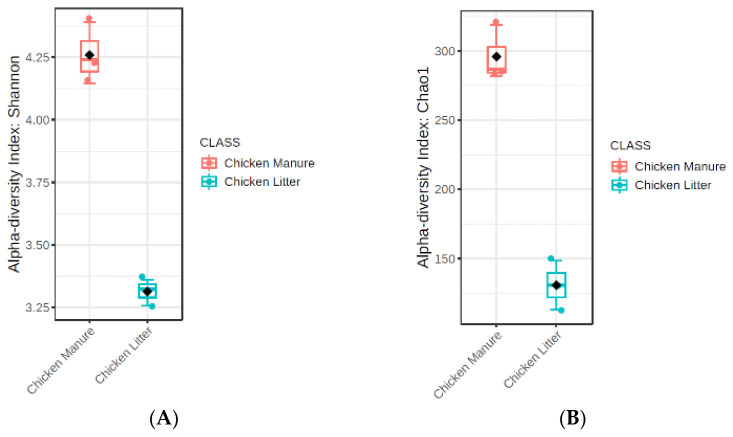
(**A**) Shannon indexes of microbial community diversity *p* ≤ 0.1; statistical analyses were performed with the Mann–Whitney test (**B**). Chao 1 indexes of microbial community richness, *p* ≤ 0.1; statistical analysis was performed with the Mann–Whitney test in collected chicken waste samples (chicken manure: CM_1, CM_2, CM_3 (orange dots) and chicken litter: CL_1, CL_2, CL_3 (blue dots)).

**Figure 4 biomolecules-12-01132-f004:**
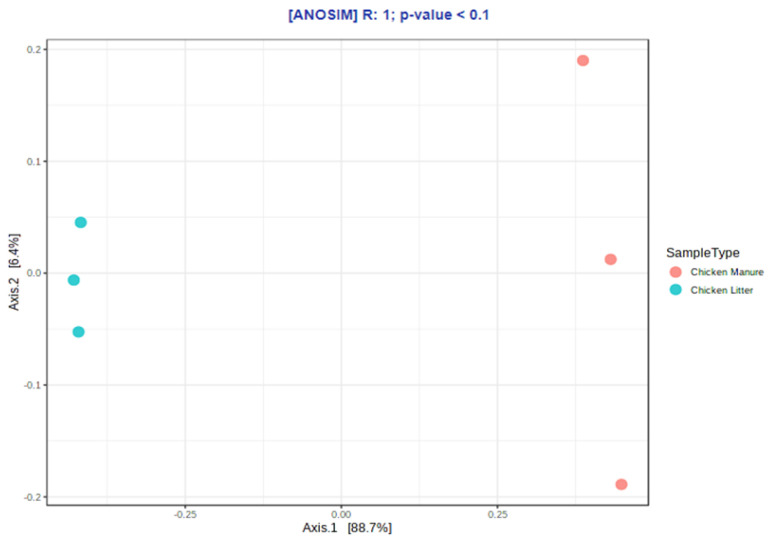
Principal coordinate analysis (PCoA) of microbial community compositions based on the Bray–Curtis dissimilarity matrices (ANOSIM) in collected chicken waste samples (chicken manure: CM_1, CM_2, CM_3 (orange dots) and chicken litter: CL_1, CL_2, CL_3 (blue dots)).

**Figure 5 biomolecules-12-01132-f005:**
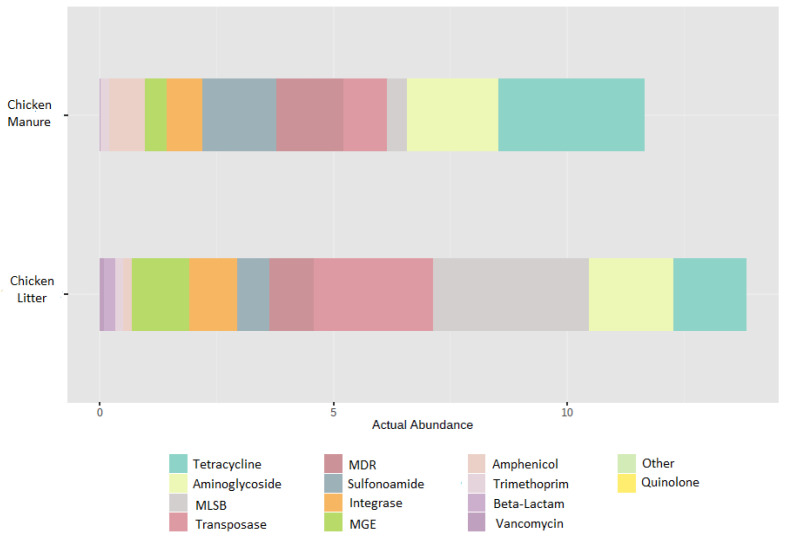
Relative abundance of the detected antibiotic-resistance gene groups, multi-drug resistance mechanisms, mobile genetic elements, integrase and transposase, visualized as gene copy number normalized per 16S rRNA gene copies. The visualization represents the mean value for three replicates.

**Figure 6 biomolecules-12-01132-f006:**
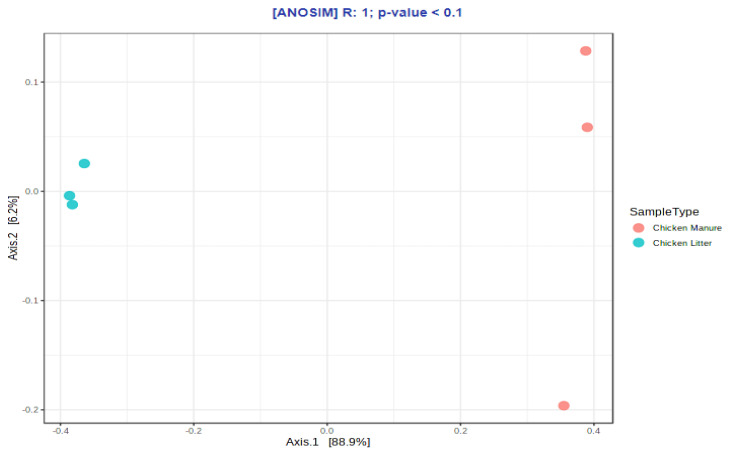
Principal coordinate analysis (PCoA) of ARG and MGE compositions in collected chicken waste samples based on Bray–Curtis dissimilarity matrices (chicken manure: CM_1, CM_2, CM_3 (orange dots), and chicken litter: CL_1, CL_2, CL_3 (blue dots)).

**Figure 7 biomolecules-12-01132-f007:**
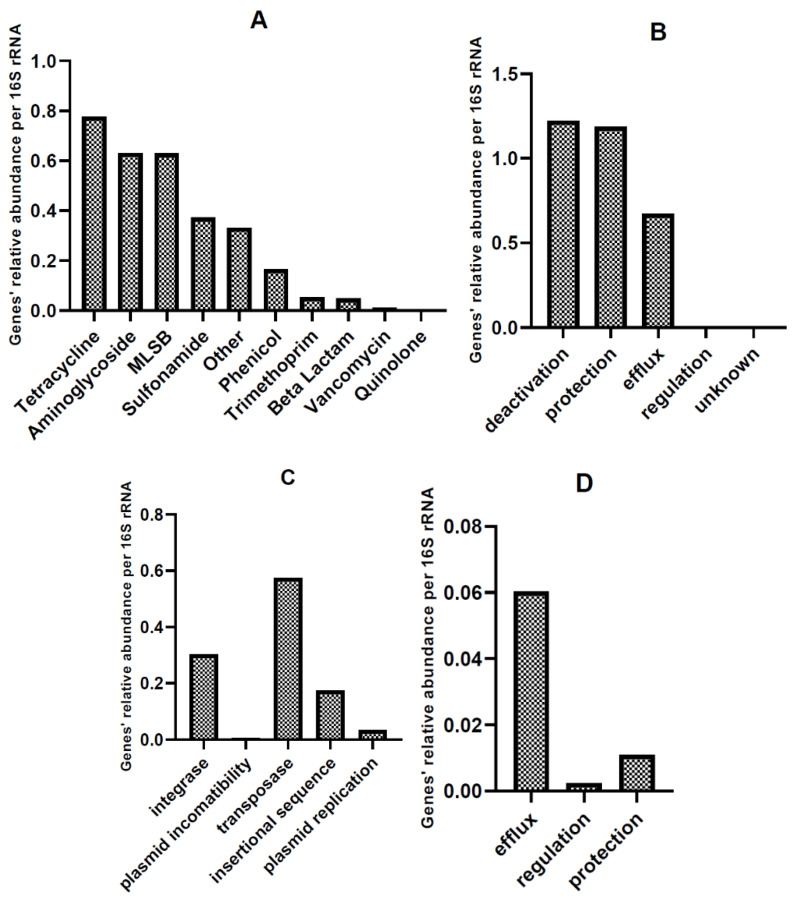
Distribution of ARGs, presented as mean values for all chicken waste samples, categorized as the following: (**A**). gene classes, (**B**). mechanisms of resistance, (**C**). mechanisms related to MGE, (**D**). mechanisms related to MDR identified in the analyzed chicken waste samples.

**Figure 8 biomolecules-12-01132-f008:**
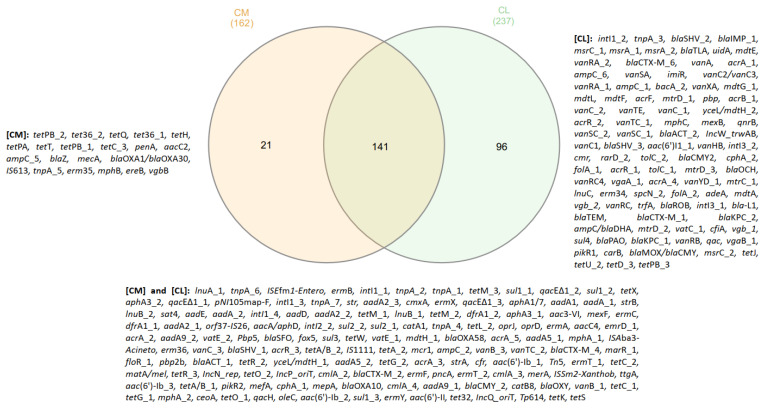
Comparison between CL and CM resistomes.

**Figure 9 biomolecules-12-01132-f009:**
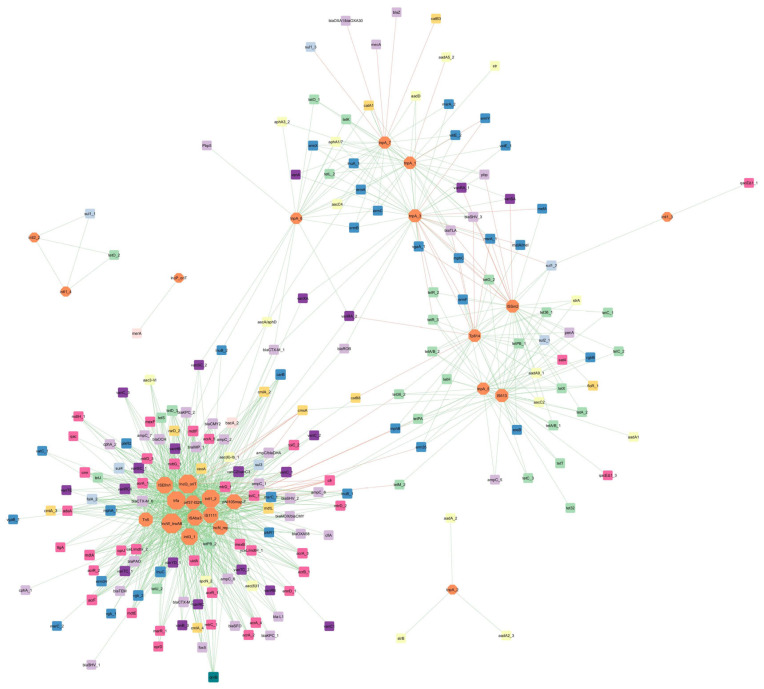
ARG and MGE interaction network analysis presented in ’organic layout’. A connection shows a strong and significant correlation based on Spearman’s rank analysis (|r|/.0.9, *p* < 0.01). The red and green edges indicated the indexes of positive and negative correlations between ARGs and MGEs, respectively. The size of the nodes show the degree of the interactions.

**Figure 10 biomolecules-12-01132-f010:**
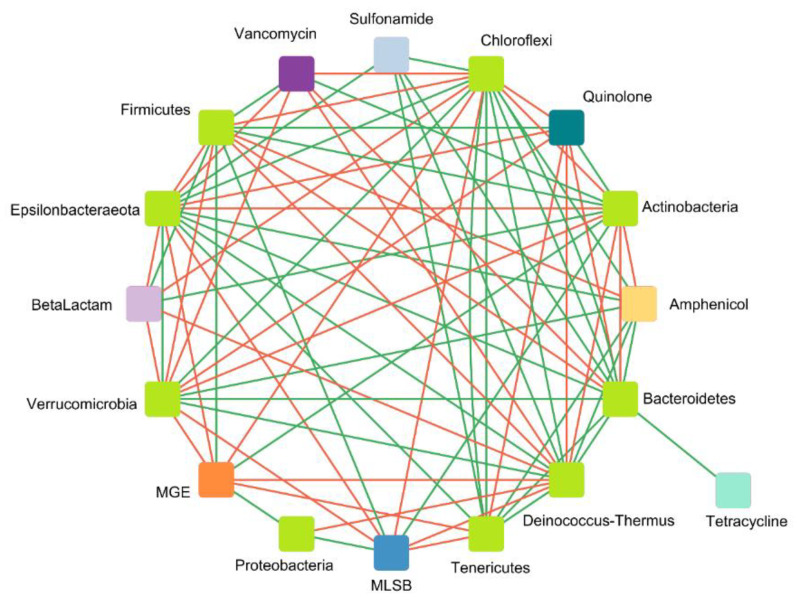
Microbial taxon and gene class interaction network analysis as a ’circular layout‘ based on the co-occurrence of ARG groups and microbial taxa at phylum level. A connection shows a significant correlation based on Spearman’s rank analysis (|r|/.0.7, *p* < 0.05). The green and red edges indicate positive and negative correlations between ARGs and taxa, respectively.

**Table 1 biomolecules-12-01132-t001:** Selective agar screening for AMR pathogens.

Bacteria Species	Screening Selective Agar + Antibiotic (Concentration) Using the EUCAST Breakpoints	Selecting for
*Klebsiella pneumoniae*	Simmons Citrate Agar + inositol Imipenem (16 mg/L)	Carbapenem resistance
*Klebsiella pneumoniae*	Simmons Citrate Agar + inositol Cefotaxime (4 mg/L)	Extended-spectrum beta-lactamase
*Escherichia coli*	Eosin Methylene Blue Agar Imipenem (16 mg/L)	Carbapenem resistance
	MacConkey Agar Imipenem (16 mg/L)	
*Escherichia coli*	Eosin Methylene Blue Agar Cefotaxime (4 mg/L)	Extended-spectrum beta-lactamase
	MacConkey Agar Cefotaxime (4 mg/L)	
*Enterococcus* sp.	Brilliance VRE Agar	Vancomycin resistance
*Acinetobacter baumannii*	CHROMagar Acinetobacter Imipenem (16 mg/L)	Carbapenem resistance
*Pseudomonas aeruginosa*	Cetrimide Agar Imipenem (16 mg/L)	Carbapenem resistance
*Staphylococcus aureus*	Mannitol salt Agar Oxacillin (4 mg/L)	Methicillin resistance
*Staphylococcus aureus*	Mannitol Salt Agar Vancomycin (4 mg/L)	Vancomycin resistance

**Table 2 biomolecules-12-01132-t002:** Physicochemical properties of the two kinds of chicken manure from Polish commercial farm and non-manured soil.

Parameter		Soil	Chicken Manure (CM)	Chicken Litter (CL)
pH		6–6.5	8	6.5
heavy metals [mg/kg]	mercury (Hg)	<0.05	<0.05	<0.05
	chrome (Cr)	5.56	<2.00	<2.00
	zinc (Zn)	19.7	616	430
	cadmium (Cd)	<0.4	0.687	0.478
	copper (Cu)	3.41	86.7	66.3
	nickel (Ni)	<3.00	6.27	9.2
	lead (Pb)	7.74	<5.00	<5.00
macronutrients [mg/kg]	calcium (Ca)	110	1140.0	6670
	magnesium (Mg)	57.7	945	613
	total phosphorus (P)	205	2430	1510
ammoniacal nitrogen (NH4-N) [mg/kg]		<0.00233	3200	2400
total nitrogen (N) [%wt]		0.0926	1.22	2.78
dry organic matter (OM) [%]		2.2	18.4	60.2

**Table 3 biomolecules-12-01132-t003:** Amount of bacteria isolated from each medium type, calculated as cfu/mL.

Medium	Antibiotic	Temperature [°C]	Chicken Waste	[cfu/mL]
Mannitol Salt Agar	Vancomycin	25	CM	3.1 × 10^6^
		25	CL	2.1 × 10^5^
		37	CM	1.8 × 10^5^
		37	CL	6.5 × 10^4^
	Oxacillin	25	CM	----
		25	CL	4.6 × 10^5^
		37	CM	5.1 × 10^6^
		37	CL	4.1 × 10^7^
Eosin Methylene Blue Agar	Cefotaxime	25	CM	7.2 × 10^8^
		25	CL	5.1 × 10^5^
		37	CM	3.9 × 10^5^
		37	CL	1.9 × 10^4^
	Imipenem	25	CM	1.2 × 10^7^
		25	CL	3.3 × 10^5^
		37	CM	----
		37	CL	1.0 × 10^3^
MacConkey Agar	Cefotaxime	25	CM	3.2 × 10^5^
		25	CL	4.1 × 10^4^
		37	CM	4.7 × 10^6^
		37	CL	3.8 × 10^5^
	Imipenem	25	CM	2.2 × 10^3^
		25	CL	1.5 × 10^2^
		37	CM	4.1 × 10^4^
		37	CL	1.5 × 10^3^
Simmons Citrate Agar	Cefotaxime	25	CM	3.3 × 10^6^
		25	CL	2.4 × 10^5^
		37	CM	1.4 × 10^2^
		37	CL	---
	Imipenem	25	CM	4.2 × 10^6^
		25	CL	----
		37	CM	6.2 × 10^6^
		37	CL	----
Cetrimide Agar	Imipenem	25	CM	1.2 × 10^4^
		25	CL	2.1 × 10^6^
		37	CM	1.1 × 10^5^
		37	CL	1.0 × 10^4^
CHROMagar™ Acinetobacter	Imipenem	25	CM	----
		25	CL	----
		37	CM	----
		37	CL	----
Brilliance VRE Agar	Vancomycin	25	CM	-----
		25	CL	2.4 × 10^3^
		37	CM	1.3 × 10^2^
		37	CL	----

CL—chicken litter; CM—chicken manure; cfu—colony forming unit.

**Table 4 biomolecules-12-01132-t004:** Antibiotic-resistance phenotypes of isolated strains from two types of chicken wastes—CL and CM.

Species	No. of Strains/All Isolates	Source	Phenotype		MDR (Yes/No)
			Resistance	Intermediate Resistance	
*Acinetobacter johnsonii*	1/1	CL	TZP	CIP(I), PRL(I)	No
*Citrobacter freundii*	1/1	CL, CM	AMC, CXM, CAZ, CN, TOB, CTX, CIP, CXM-AK	TZP(I)	Yes
*Escherichia coli*	1/23	CL	AMC, CXM, CAZ, CN, TOB, CTX, CIP, CXM-AK	IMP(I)	Yes
	4/23	CL, CM	AMC, TZP, CXM, CAZ, CN, TOB, CTX, CIP, CXM-AK		Yes
	8/23	CL	AMC, CXM, CAZ, CN, TOB, CTX, CIP, CXM-AK	TZP(I)	Yes
	1/23	CL	AMC, CXM, CAZ, CN, TOB, CTX, FEP, CXM-AK	TZP(I)	Yes
	8/23	CL, CM	AMC, CXM, CAZ, CN, TOB, CTX, CIP, CXM-AK		Yes
	1/23	CL	AMC, CXM, CAZ, MEM, CN, TOB, CTX, CIP, CXM-AK	TZP(I)	Yes
*Klebsiella aerogenes* (*Enterobacter aerogenes*)	1/3	CL		AMC(I)	No
	2/3	CL	AMC	CXM(I)	No
*Myroides odoratimimus*	2/13	CL, CM	TZP, CN, AK, TOB, CIP, SXT, ATM, LEV		Yes
	1/13	CL, CM	CN, AK, TOB, CIP, SXT, PRL, ATM, LEV	TZP(I), IMP(I)	Yes
	2/13	CL, CM	CN, AK, TOB, CIP, SXT, PRL, ATM, LEV	TZP(I)	Yes
	1/13	CL, CM	CN, AK, TOB, SXT, PRL, ATM	TZP(I)	Yes
	2/13	CL, CM	CN, AK, TOB, CIP, SXT, PRL, ATM, LEV		Yes
	1/13	CL, CM	TZP, CN, AK, TOB, SXT, PRL, ATM	IMP(I), CIP(I)	Yes
	1/13	CL, CM	CN, TOB, CIP, SXT, PRL, ATM, LEV	TZP(I), AK(I)	Yes
	2/13	CL, CM	CN, AK, TOB, SXT, PRL, ATM	TZP(I), CIP(I)	Yes
*Myroides odoratus*	2/2	CM	TZP, IMP, MEM, CN, AK, TOB, CIP, SXT, PRL, ATM, LEV		Yes
*Providencia rettgeri*	1/3	CM	AMC, CIP, TGC, CT	CXM(I), IMP(I)	Yes
	1/3	CM	AMC, TGC, CT, SXT	CXM(I), IMP(I), CTX(I)	Yes
	1/3	CM	AMC, TZP, CIP, TGC	IMP(I), FEP(I)	Yes
*Serratia marcescens*	1/5	CL	AMC, CXM, CIP, CXM-AK, CT	TGC(I)	Yes
	3/5	CL	AMC, CXM, CIP, CXM-AK, CT		Yes
	1/5	CL	AMC, CXM, CXM-AK		Yes
*Staphylococcus lentus*	2/19	CL	CIP, LEV, E, TE, DA	STX(I)	Yes
	2/19	CL	CN, CIP, STX, LEV, E, TE, DA		Yes
	3/19	CL	CIP, LEV, OB, TE, DA		Yes
	2/19	CL	CN, CIP, STX, LEV, LZD, E, TE, DA		Yes
	2/19	CL	TE	CIP(I), LEV(I), DA(I)	Yes
	2/19	CL	TE	CIP(I), LEV(I)	Yes
	1/19	CL	CN, CIP, SXT, LEV, OB, E, TE, DA		Yes
	1/19	CL	CIP, TGC, LEV, LZD, E, TE, DA		Yes
	1/19	CL	CN, CIP, SXT, LEV, E, TE, DA		Yes
	1/19	CL	CIP, TGC, LEV, LZD, TE, DA		Yes
	1/19	CL	CIP, LEV, ER, TET, DA		Yes
	1/19	CL	CIP, SXT, LEV, ER, TET, DA		Yes

AMC—amoxicillin/clavulanic acid, AK—amikacin, ATM—aztreonam, CXM—cefuroxime, CN—gentamicin, CIP—ciprofloxacin, E—erythromycin, TE—tetracycline, TGC—tigecycline, TZP—piperacillin/tazobactam, LEV—levofloxacin, CAZ—ceftazidime, IMP—imipenem, MEM—meropenem, TOB—tobramycin, CTX—cefotaxime, FEP—cefepime, CXM-AK—cefuroxime-axetil, LZD—linezolid, DA—clindamycin, CT—colistin, STX—trimethoprim/sulfamethoxazole, PRL—piperacillin; (I) intermediate susceptibility according to EUCAST.

**Table 5 biomolecules-12-01132-t005:** The ARGs found in the two studied types of chicken waste (CL and CM), expressed as relative abundance; values for CM and CL are presented as mean values; relative gene abundance is calculated per 16S rRNA gene copies. The data are sorted from most to least abundant.

Gene Group	CL (Mean)	CM (Mean)	Chicken Waste (Mean)
MGE	1.2556307984	0.3211213793	0.7883760889
Tetracycline	0.5144201057	1.0377003019	0.7760602038
Aminoglycoside	0.6050128022	0.6581070783	0.6315599403
MLSB	1.1137056246	0.1455982502	0.6296519374
Sulfonamide	0.2256385199	0.5193078503	0.3724731851
Other	0.2076692490	0.4556390180	0.3316541335
Integrons	0.3429847709	0.2639967037	0.3034907373
Phenicol	0.0691178113	0.2602892051	0.1647035082
MDR	0.1157465734	0.0317774276	0.0737620005
Trimethoprim	0.0476679473	0.0578028990	0.0527354232
Beta Lactam	0.0909582761	0.0055880528	0.0482731645
Vancomycin	0.0196980994	0.0006651844	0.0101816419
Quinolone	0.0006499213	0.0000000000	0.0003249607

**Table 6 biomolecules-12-01132-t006:** The most prevalent ARGs found in CM; relative gene abundance calculated per 16S rRNA gene copies.

Gene	CM (Mean)
*tet*PB_2	0.2751246220
*tnp*A_2	0.2459280497
*cmx*A	0.2216566087
*qac*E∆1_1	0.2021822017
*tet*M_3	0.1993868087
*tet*X	0.1452556967
*qac*E∆1_2	0.1434562960
*tet*M_1	0.1427971303
*sul*2_2	0.1301656963
*sul*1_1	0.1214871360

**Table 7 biomolecules-12-01132-t007:** The most prevalent ARGs found in CL; relative gene abundance calculated per 16S rRNA gene copies.

Gene	CL (Mean)
*lnu*A_1	0.6792154677
*tnp*A_6	0.4736300717
ISE*fm1-Entero*	0.3201372400
*erm*B	0.2429636293
*tet*M_3	0.1942361840
*int*I1_1	0.1446150687
*tnp*A_2	0.1364071333
*tnp*A_1	0.1259366313
*tet*M_1	0.1186746973
*sul*1_1	0.1135728363

## Data Availability

Datasets used during the study have been shared as Appendix A.

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
