# Peer review of "A Comprehensive Study of the Microbiome, Resistome, and Physical and Chemical Characteristics of Chicken Waste from Intensive Farms"

_biomolecules, 2022, doi:10.3390/biom12081132_

Round 1

Reviewer 1 Report

In the manuscript entitled “A comprehensive study of the microbiome and resistome of chicken waste from intensive farm" authored by  Aleksandra Błażejewska et al., the authors present in an eloquent and clear manner the results of the study.

The manuscript does not require editing of English language and style.

The methods and the protocols are standardized and suitable for the study aim. The techniques used to identify and characterize antimicrobial resistance involve both classical and advanced tools.

The novelty of the research are well underlined.

Still, there is a recommendation regarding the whole manuscript (introduction, discussion and conclusion). To highlight the importance of the results, One Health concept should be included. The authors should underline the value of the obtained results using epidemiological connections.

Author Response

Dear Reviewer 1

Authors thanks to Reviewer for their valuable opinions and time spend for our manuscript. The answer for Reviewers recommendation are listed below (marked in red). All correction done in the main manuscript are also marked in red.

The sections describing the One Health approach have been added to the 'Introduction' and 'Results and discussion' sections.

Reviewer 2 Report

This is an interesting study on the microbiome and resistome of chicken waste from the intensive farm by Blazejewska et al. However, some clarifications are needed to proceed with the present research.

2.1. Manure sample. What are those different sources of chicken waste samples? Chicken farms? Not clear.

The authors do not provide any details on sampling, this must be included in the present work.

Line 140. Luria Bertani broth. Which microorganisms was the present media used for?

Line 142. Correct to in the power of one, two, etc.

Line 143-144. Why were those microorganisms selected for isolation? At least the microorganisms of interest should be added to the main text of the manuscript or even Table S2 could be moved to the main text. An additional table with numbers of isolates or positive samples obtained for each microbial group may be added.

Line 151. Could the authors mention how many colonies were selected per one inoculated and incubated agar?

Line 156. Had the AMR testing been initiated before the microbial identification?

Line 160-162. There might be differences in AMR between the individual strains, you can calculate how many strains were resistant vs. Susceptible or calculate sd for inhibition zone but not exclude those strains from the experiment. There are wild strains therefore difficult to imagine how the authors can get repetitive results from those.

Line 179-183. I do not understand the criteria for AMR detection by Vitek and disk diffusion methods.

I do not think that is correct to treat the strains as resistant based only on the absence of an inhibition zone. The authors must state that the criteria are not established and could not be evaluated, this must be followed throughout the manuscript.

Line 212. S4? I do not think that this is a good practice to use the information from the supplementary file to describe the main results. At least the most important genes should be moved into the main text. Were all the mentioned primers designed for the present study needs? Add the appropriate reference(s) if not.

Line 217. The authors should separate the data analysis from the this and results and discussion sections.

Line 255 and further in the manuscript. Please do not rewrite the previous studies, only stress your findings in context with the previous research.

Line 295. What are potentially repetitive strains?

Figure 1. Am I right that 9+7 were microbial species isolated? Not clear. The authors also need to define which microprograms are opportunistic pathogens and which are not.

Line 312 and elsewhere in the manuscript. Please show the numbers of positive strains alongside with %

Table 2. Please separate intermediately resistant strains in the separate column, no. of MDR strains also could be indicated.

Another question – Had the resistance in those strains been confirmed with ARGs analysis?

Lines 349-384. Please do not rewrite the previous studies + why only the studies conducted in Africa are mentioned in context with the research performed in the EU?

Line 393. “Many potentially pathogenic bacteria”. The present report does not show that. You can use the term “opportunistic pathogens” but they need to be defined in the manuscript.

L471 -513 The authors have already described the effects of CM and CL on microbiota, so could this part could be reduced to show only the most important findings? Also L497- 522 could be significantly reduced.

3.2.4.1. subsection only contains the description of the previous studies without any interesting reference to the present results

Figure 5. Please provide the explanations for the abbreviations

Line 562. How was the correspondence of the ARG and the respective antibiotics detected? Have been some analyses conducted on the farm?

Figure 7. In my mind, would be correct to analyze separately the distribution of the ARGs, then mono and multi-resistance, then according to the resistance mechanisms. This is all mixed in the Figure 7 at the present. + legend should be self-explanatory.

Line 587. Units for 0.07?

Figure 8. The legend should be self-explanatory. Difficult to interpret the results which are shown in the Figure 8. A proportion of microbiota is naturally resistant to antibiotics hence is difficult to understand the results without any reference to microbial species.   

3.2.4.2. I am cautious with this section because epidemiological evidence including microbial species should be taken into account for data analysis. As I stated before, a part of microbiota is naturally resistant to various antibiotics and this needs to be taken into context. The figure is of very bad quality.

Line 665. “prophylaxis in animal”. Please specify.

Line 675. “administered drug is excreted”. Which drugs if the authors describe ARGs?

Line 737-746. Could the authors consider the addition of i.e. “waste physically-chemical characteristics” to the title? Otherwise, this part of the conclusion seems to be out of the present study.

Table S1. Add the numbers of strains. Replace “Intermediate strains” with “Intermediate resistant strains”.  

Figure S1. Add the legend.

Author Response

Dear Reviewer 2

Authors thanks to Reviewer for their valuable opinions and time spend for our manuscript. The answer for Reviewers recommendation are listed in the attachment (marked in red). All correction done in the main manuscript are also marked in red.

Reviewer 3 Report

This reviewer enjoyed reading this well-written manuscript. The topic is very interesting, and the conclusion is supported by the performed assays. 

The main concern I have is related to the sampling: The samples were collected from CM (chicken manure from 125 laying hens) and CL (chicken litter from broilers). There are several pitfalls in this design: 

1) You cannot make a solid and meaningful decision from CM of one chicken population, and CL from another population;

2) How did you sample the manure and litter? Did you get manure from a single chicken? Did you obtain litter from a single area? How would the used manure and litter represent the whole situation in the chicken populations? More importantly, what was the rationale that you collected only two samples for this work (one CM; one CL)? Would the conclusion made from these two samples be meaningful to represent the situation in Poland?

3) With two samples, a meaningful statistical analysis cannot be performed. You did not mention statistical analysis in Results, but not in M & M.

Author Response

Dear Reviewer 3

Authors thanks to Reviewer for their valuable opinions and time spend for our manuscript. The answer for Reviewers recommendation are listed in the attachment (marked in red). All correction done in the main manuscript are also marked in red.

Reviewer 4 Report

The research investigates an utmost important subject, that of AMR in the environment niche, one of the most important (and sometimes neglected) source for antimicrobial resistance mobility. The authors attempt a complex view on the AMR, combining the investigation on metagenome, resistome and antibiotic fingerprints of bacteria in chicken manure and litter, with the analysis on the fertiliser quality of the manure/litter, almost looking into a "cost/benefit ratio" of using these by products in agriculture. 

It would have been interesting to see a comparison between the same type of litter/manure from each category of birds since the economic life of the category (broilers and layers, respectively) is different, thus the moment of sampling could be important (“Chicken waste samples were collected from two different sources of meat production at a large, commercial, meat-producing farm in Poland (CM - chicken manure from laying hens and CL - chicken litter from broilers” – sampling moment during the economic life not specified). (“Chicken waste samples were collected from two different sources of meat production at a large, commercial, meat-producing farm in Poland (CM - chicken manure from laying hens and CL - chicken litter from broilers)” – the moment of sampling during the economic life is not specified; the hens, as they are kept for a production cycle of 70 weeks as opposed to broilers -45 days, are more likely to be exposed to antibiotic resistant bacteria, even if only due to their longer life.

No provenance is indicated for the soil which was analysed. Did it come from the proximity of the chicken farms?

It is known that antimicrobial resistance is increased in some bacteria in the presence of heavy metals; there is no discussion of the results from this point of view. The only mention in the respect of the level of heavy metals is “Our findings indicate that the examined chicken waste can be a sufficient source of nutrients essential for plant growth like nitrogen and phosphorus, especially considering that the amounts of heavy metals in the chicken manures were within acceptable levels and would not likely pose an environmental threat when directly applied to the soil.” Still, it might influence the “alarming diversity and abundance of ARGs and MGEs, and our data indicates positive correlations between them suggesting a considerable risk of ARG spread into the environment and their transfer between bacterial species.” – as the authors say. 

No provenance is indicated for the soil which was analysed. Did it come from the proximity of the chicken farms?

It is known that antimicrobial resistance is increased in some bacteria in the presence of heavy metals; there is no discussion of the results from this point of view. The only mention in the respect of the level of heavy metals is “Our findings indicate that the examined chicken waste can be a sufficient source of nutrients essential for plant growth like nitrogen and phosphorus, especially considering that the amounts of heavy metals in the chicken manures were within acceptable levels and would not likely pose an environmental threat when directly applied to the soil.” Still, it might influence the “alarming diversity and abundance of ARGs and MGEs, and our data indicates positive correlations between them suggesting a considerable risk of ARG spread into the environment and their transfer between bacterial species.” – as the authors say.

Author Response

Dear Reviewer 4

Authors thanks to Reviewer for their valuable opinions and time spend for our manuscript. The answer for Reviewers recommendation are listed in the attachment (marked in red). All correction done in the main manuscript are also marked in red.

Round 2

Reviewer 2 Report

Dear Authors,

Thank you for your responses according points have been raised.

I have a few additional suggestions for improvement of the manuscript.

Insert “Poland” to keywords.

Line 348. “amount of bacteria”. Consider replacement with “bacterial counts”

Line 468-474. Omit this part – too long and contain general knowledge addressed by WHO.

Line 479-480. Is this description needed?

Figure 9 of higher resolution could be added also as e.g. graphical abstract alongside with the supplementary material. Those important results are not visualized enough at the present moment.

Supplementary files. Heatmap on ARG - A figure legend is a text which provides the readers with the information necessary to understand the figure without reading the main text. Titles are missing for supplementary files No.1 and No. 2.

Author Response

Dear Authors,

- Thank you for your responses according points have been raised. I have a few additional suggestions for improvement of the manuscript.

The authors thank the Reviewer for his valuable opinion and time. The manuscript has been sent once again to a native speaker for correction.

- Insert “Poland” to keywords.

The keyword has been added.

- Line 348. “amount of bacteria”. Consider replacement with “bacterial counts”

The change has been made.

- Line 468-474. Omit this part – too long and contain general knowledge addressed by WHO.

This part has been removed from the text.

- Line 479-480. Is this description needed?

This part has also been removed from the test.

- Figure 9 of higher resolution could be added also as e.g. graphical abstract alongside with the supplementary material. Those important results are not visualized enough at the present moment.

Figure 9 has been submitted additionally submitted as graphical abstract.

- Supplementary files. Heatmap on ARG - A figure legend is a text which provides the readers with the information necessary to understand the figure without reading the main text. Titles are missing for supplementary files No.1 and No. 2.

Supplementary materials have been submitted once again with titles.

Reviewer 4 Report

By adding supplementary explanations, clarification of some of the issues and further discussions, the authors have now improved the quality of the paper  and it will help further improvement of the research in the field.

Author Response

Dear Reviewer

- By adding supplementary explanations, clarification of some of the issues and further discussions, the authors have now improved the quality of the paper  and it will help further improvement of the research in the field.

The authors thank the Reviewer for his valuable opinion and time. The additional manuscript English language and style correction has been made by native speaker.